# PhyCAGE: Physically Constrained Compositional 3D Asset Generation from a Single Image

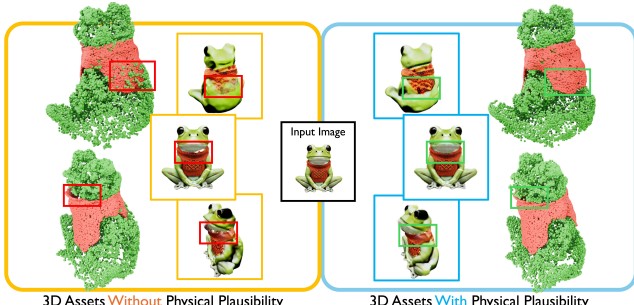

Figure 1: PhyCAGE can generate compositional 3D assets with interactive objects in a physically plausible manner. The generated 3D Gaussian Splatting shows better visual performance and physical plausibility under Material Point Method (MPM) simulation.

## Abstract

We present **PhyCAGE**, the first approach for **Phy**sically constrained **C**ompositional 3D **A**sset **GE**neration from a single image. Given an input image, we first generate consistent multi-view images for components of the assets. These images are then fitted with 3D Gaussian Splatting representations. To ensure that the Gaussians representing objects are physically compatible with each other, we introduce a Physical Simulation-Enhanced Score Distillation Sampling (PSE-SDS) technique to further optimize the positions of the Gaussians. It is achieved by setting the gradient of the SDS loss as the initial velocity of the physical simulation, allowing the simulator to act as a physics-guided optimizer that progressively corrects the Gaussians' positions to a physically compatible state. Experimental results demonstrate that the proposed method can generate physically plausible compositional 3D assets given a single image.

## 1 Introduction

Generating 3D shapes conditioned on 2D image input lies at the core of many applications, such as virtual reality (VR), augmented reality (AR), video gaming, and robotics. Recently, this field has seen remarkable progress, thanks to advancements in AI techniques, including transformers (Vaswani et al., 2017) and diffusion models (Ho et al., 2020).

While existing methods (Hong et al., 2023; Wang et al., 2023b; Shi et al., 2023; Liu et al., 2023b; Tang et al., 2023) mainly focus on the image-to-3D generation of a single object, this paper explores the more intricate challenge of generating compositional 3D assets: when presented with an image of an asset containing two compositional objects, our goal is to generate separate 3D representations of each component while ensuring that their relationships are semantically coherent and geometrically and physically plausible.

A simple strategy is to generate the entire assets as a holistic 3D mesh and subsequently use surface segmentation to separate the individual objects, as implemented in Part123 (Liu et al., 2024a)

and SAMPart3D (Yang et al., 2024). However, mesh segmentation usually leads to incomplete surfaces and disregards the relationships among objects. Alternative methods involve generating each component as an individual object and then combining them into a single model using estimated spatial placement, such as the similarity transformation that includes translation, rotation, and scaling. Examples of this approach can be found in (Epstein et al., 2024; Chen et al., 2024a). However, they struggle to manage complex spatial relationships that extend beyond simple similarity transformations. They fail in situations where non-rigid object deformation is required and often result in shape penetrations.

We observe that physical information, such as supporting relationships, stability, and affordance, can offer valuable clues for generating the shapes of interactive objects. For instance, objects in static scenes should exhibit stability. In a scene depicting "a frog wearing a sweater", the frog should possess adequate body structure to support the sweater; otherwise, gravity will cause the sweater to fall off. To this end, we integrate differentiable physical simulations into the process of compositional 3D asset generation.

Specifically, given an input image, we generate consistent multi-view images for both the entire assets, a foreground component, and an inpainted occluded background. The multi-view images are subsequently fitted with 3D Gaussian Splatting (Kerbl et al., 2023) representations. Then, to ensure the physical plausibility of the assets, we build upon the Score Distillation Sampling (SDS) (Poole et al., 2022) method and introduce a physical simulation-enhanced SDS to further optimize the geometry (i.e., positions of Gaussians) for the objects. To ensure visual consistency with the input image, we incorporate image loss, i.e., the difference between the input image and rendered image from the generated object as a complement.

We observe that directly applying the SDS and image loss gradient to update Gaussians' positions results in penetrations and non-physical artifacts. Our proposed physical simulation-enhanced SDS delegates updates of Gaussians' positions to the physical simulation instead of the optimizer in the training process. By setting the loss gradient as the initial velocity of the physical simulation, the simulator serves as a physics-guided optimizer, which progressively corrects the particle positions by solving the physical system.

Experiments demonstrate the proposed method can generate physically plausible compositional 3D assets given a single image. Our contributions are as follows:

- We design a novel pipeline for image-based compositional 3D asset generation, particularly focusing on interactive objects with strong spatial coupling.

- We propose a physical simulation-enhanced Score Distillation Sampling to optimize 3D Gaussians in a physically plausible manner.

- We are the first to mitigate inter-component penetration when generate 3D compositional assets from a single image, facilitating downstream applications.

## 2 RELATED WORK

### 2.1 IMAGE CONDITIONED 3D GENERATION

With the remarkable success of diffusion models (Sohl-Dickstein et al., 2015) in the 2D domain (Ho et al., 2020; Rombach et al., 2022), numerous studies have started investigating how to build 3D generation models. One approach involves generating 3D assets by distilling knowledge from pre-trained 2D generators (Poole et al., 2022; Qian et al., 2023). DreamFusion (Poole et al., 2022) proposed Score Distillation Sampling (SDS) to optimize a NeRF (Mildenhall et al., 2020) model with images generated by a 2D generator. Meanwhile, Magic123 (Qian et al., 2023) employed a coarse-to-fine strategy to enhance both the speed and quality of the generated models. The other technical solution involves directly training 3D generators using ground-truth 3D data, and training denoising models to produce 3D shapes from image conditions. Notable works include (Wang et al., 2023b; Zheng et al., 2023; Zhang et al., 2023a; 2024). LRM (Hong et al., 2023) reformulated 3D generation as a deterministic 2D-to-3D reconstruction problem. Synthesizing multi-view consistent images enhances the capabilities of 3D generation or reconstruction, as shown in Zero123++ (Shi et al., 2023) and Syncdreamer (Liu et al., 2023b).

The aforementioned approach generates 3D data in the form of a single, entangled representation, which is not ideal for numerous downstream applications that require semantically compositional shapes.

## 2.2 Compositional 3D Reconstruction and Generation

ObjectSDF (Wu et al., 2022) and ObjectSDF++ (Wu et al., 2023) introduced an object-composition neural implicit representation, which allows separate reconstruction of each piece of furniture within a room, solely based on image inputs. DELTA (Feng et al., 2023) presented hybrid explicit-implicit 3D representations, designed for the joint reconstruction of compositional avatars. This includes the integration of components such as the face and body, or hair and clothing, respectively. Similar compositional avatars generation with the SMPL (Loper et al., 2015) body pror works can be found in (Hu et al., 2023; Dong et al., 2024; Wang et al., 2023a;c). AssetField (Xiangli et al., 2023) proposed to learn a set of object-aware ground feature planes to represent the scene and various manipulations could be performed to rearrange the objects. (Po & Wetzstein, 2023; Cohen-Bar et al., 2023) jointly optimized multiple NeRFs, each for a distinct object, over semantic parts defined by text prompts and bounding boxes. (Epstein et al., 2024) and SceneWiz3D (Zhang et al., 2023b) eliminated the requirements for user-defined bounding boxes by simultaneously learning the layouts. Since the text could be problematically complicated when describing complex scenes, GraphDreamer (Gao et al., 2024) used scene graphs as input instead. Frankenstein (Yan et al., 2024) extended 3D diffusion approach for building a compositional scene generation tool.

In this paper, we adhere to the SDS-based method but incorporate physics simulation to address the inherent ambiguity of the 2D-to-3D problem and enhance the physical plausibility of the 3D assets.

## 2.3 Physics Based 3D Generation

Several attempts have been made to generate physically compatible objects. Aiming to generate physically compatible objects, (Chen et al., 2024b) proposed an SDS-based method with rigid-body simulation, which can generate self-supporting objects from text. (Guo et al., 2024) presented a method of generating objects constrained by static equilibrium from a single image. In addition to object geometry generation from texts or images, there are existing works focusing on learning the objects' internal material parameters. In (Zhang et al., 2025), an approach was proposed to distill dynamic priors from pre-trained video diffusion models by minimizing the discrepancy between physical simulation and diffusion-generated videos. (Liu et al., 2024b) further utilized a more complex viscoelastic material model to simulate the objects and optimize the physical parameters via SDS. The above methods mainly focus on a single object, approaches are proposed for physically plausible scene reconstruction (Ni et al., 2024), language-grounded physics-based scene editing (Qiu et al., 2024). The existing methods above mainly focus on either single-object generation or rigid-body scene generation. In our work, we propose a novel approach for non-rigid compositional asset generation.

# 3 Preliminaries

## 3.1 Gaussian Splatting

3D Gaussian Splatting (Kerbl et al., 2023) (GS) has been proven efficient in 3D reconstruction tasks, due to its high inference speed and rendering quality.

Specifically, 3DGS represents 3D scenes as $N$ Gaussians with attributes $G = \{\mu_i, \Sigma_i, q_i, \alpha_i, c_i\}_{i=1}^N$, where $\mu \in \mathbb{R}^3$ is the center, $\Sigma \in \mathbb{R}^3$ is the scaling factor, $q \in \mathbb{R}^4$ is the rotation quaternion, $\alpha \in \mathbb{R}$ is the opacity value, and $c \in \mathbb{R}^3$ is the color feature.

To render an image, all Gaussians are first projected onto an image plane. Then, volumetric rendering is performed for each pixel in front-to-back depth order to produce the alpha and color maps $A_{rd}, I_{rd}$.

We use the following loss function to optimize the Gaussians:

$$\mathcal{L} = (1 - \lambda_1)\mathcal{L}_1(I_{gt}, I_{rd}) + \lambda_1\mathcal{L}_{SSIM}(I_{gt}, I_{rd}) + \lambda_2 A_{rd}(1 - A_{gt}), \quad (1)$$

where $I_{gt}$ and $A_{gt}$ are ground-truth image and mask map, $\mathcal{L}_1$ is the L1 loss function, $\mathcal{L}_{SSIM}$ is the structure similarity loss function, and $\lambda_{1,2}$ are the weighting factors.

Given a set of images $\{I_{gt,i}\}_{i=1}^{M}$, we can train 3DGS:

$$G = \text{GaussianSplatting}(\{I_{\text{gt},i}\}_{i=1}^{M}), \tag{2}$$

where we eliminate the need for ground-truth mask maps since they can be extracted from images using background removal model [1].

## 3.2 Physical Simulation

**Continuum Mechanics.** The motion of material is described by a mapping $\mathbf{x} = \phi(\mathbf{X}, t)$ from rest material space $\mathbf{X}$ to a deformed space $\mathbf{x}$ at time $t$. The Jacobian of the mapping $\mathbf{F} = \frac{\partial \phi}{\partial \mathbf{X}}(\mathbf{X}, t)$, i.e., deformation gradient measures the local rotation and strain (Bonet & Wood, 1997). Given the conservation of momentum and conservation of mass, the governing equations for describing the dynamics of an object are as follows:

$$\rho \frac{D\mathbf{v}}{Dt} = \nabla \cdot \boldsymbol{\sigma} + \mathbf{f}, \quad \frac{D\rho}{Dt} + \rho \nabla \cdot \mathbf{v} = 0, \tag{3}$$

where $\mathbf{f}$ denotes an external force, $\boldsymbol{\sigma}$ is the internal stress, the $\mathbf{v}$ and $\rho$ denote the velocity and density respectively.

**Material Point Method.** The Material Point Method (MPM) is a framework for multi-physics simulation. It utilizes the strengths of both Eulerian grids and Lagrangian particles which enables it to simulate phenomena with large deformation, topology changes, and frictional contacts. It is widely adopted for the simulation of a broad range of materials such as elastic objects, snow, sand, and cloth (Ram et al., 2015; Jiang et al., 2015; 2017; Hu et al., 2018; Fang et al., 2019). Gaussian splatting provides a particle-based explicit 3D representation, which is naturally suitable for serving as the spatial discretization of objects in physical simulation. Following (Xie et al., 2024), we run MPM on these particles directly. The MPM pipeline consists of three stages in general: particle-to-grid (P2G), grid-operation and grid-to-particle (G2P). In the P2G stage, the MPM transfers mass and momentum from particles to grids:

$$m_i^n = \Sigma_p w_{ip}^n m_p \tag{4}$$

$$m_i^n \mathbf{v}_i^n = \Sigma_p w_{ip}^n m_p (\mathbf{v}_p^n + C_p^n (\mathbf{x}_i - \mathbf{x}_p^n)), \tag{5}$$

where $p$ and $i$ denote the Lagrangian particles and Eulerian grid respectively. The term $w_{ip}^n$ denotes the B-spline basis function defined on the i-th grid, evaluated at the point $\mathbf{x}_p^n$. The particles carry properties including position $\mathbf{x}_p^n$, velocity $\mathbf{v}_p^n$, local velocity gradient $C_p^n$ and mass $m_p$ at timestep $n$. The grids are updated after the P2G stage:

$$\mathbf{v}_i^{n+1} = \mathbf{v}_i^n - \frac{\Delta t}{m_i} \sum_p \tau_p^n \nabla w_{ip}^n V_p^0 + \Delta t \cdot a, \tag{6}$$

where $a$ denotes the acceleration caused by external forces and $\tau$ denotes the stress tensor,. The updated velocities are transferred back to the particles as well as updating the positions:

$$\mathbf{v}_p^{n+1} = \sum_i \mathcal{N}(\mathbf{x}_i - \mathbf{x}_p^n) \mathbf{v}_i^n \tag{7}$$

$$\mathbf{x}_p^{n+1} = \mathbf{x}_p^n + \Delta t \mathbf{v}_p^{n+1}, \tag{8}$$

where $\mathcal{N}(\cdot)$ is the B-spline interpolation function. We utilize the MPM to simulate the interactions of compositional objects in the assets.

## 4 Method

Given an image $I \in \mathbb{R}^{H \times W}$ of an asset with two compositional objects $\{O_1, O_2\}$ described by text prompts $\tau_1$ and $\tau_2$, we would like to reconstruct a 3D representation of the two objects individually.

---

[1]https://github.com/OPHoperHPO/image-background-remove-tool

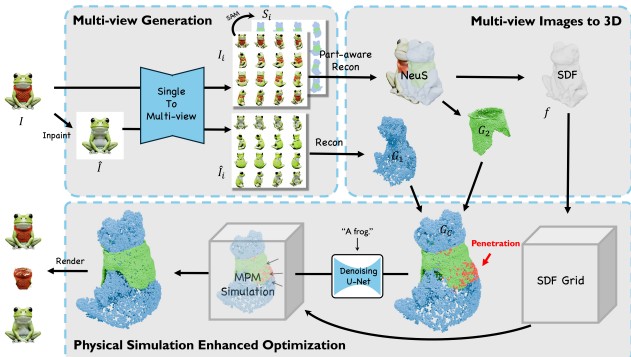

Figure 2: **The overview of PhyCAGE.** Given an input image, we first generate consistent multi-view images for the components of the assets (see Sec. 4.1). Then, we fit multi-view images with 3D Gaussian Splatting representations (see Sec. 4.2). Finally, we introduce a Physical Simulation-Enhanced SDS to further optimize the positions of the Gaussians (see Sec. 4.3).

Here we denote $O_1$ and $O_2$ as background and foreground objects respectively. We segment the foreground object in image space using Grounded-SAM (Kirillov et al., 2023) to obtain a semantic map. The image after segmentation is inpainted to complete the background object. For reconstruction, the multi-view images and the inpainted background images are generated using SyncDreamer (Liu et al., 2023b). We then reconstruct two Gaussian Splatting representations for background and foreground objects, denoted as $G_1$ and $G_2$. A physical simulation-enhanced Score Distillation Sampling (SDS) is then applied to optimize the Gaussians for obtaining a physically plausible representation.

## 4.1 MULTI-VIEW GENERATION

To reconstruct the object described in the image, we generate the multi-view images from $I$. **First**, we use Grounded-SAM (Kirillov et al., 2023; Liu et al., 2023a; Ren et al., 2024) to segment out the masks of both objects:

$$\{M_1, M_2\} = \text{GroundedSAM}(I; \tau_1, \tau_2), \tag{9}$$

where $M_1, M_2 \in \mathbb{R}^{H \times W}$. **Second**, suppose that $O_1$ is occluded by $O_2$, we use inpainting model (Rombach et al., 2022) to complete the image of $O_1$:

$$\hat{I} = \text{Inpainting}(I, M_2; \tau_1), \tag{10}$$

where $\hat{I} \in \mathbb{R}^{H \times W}$ is the inpainted image. **Third**, we use Multi-view Image Generation model (MIG) (Liu et al., 2023b; Long et al., 2024) to generate images in $W$ different views from $I$ and $\hat{I}$:

$$\{I_i\}_{i=1}^{W} = \text{MIG}(I), \quad \{\hat{I}_i\}_{i=1}^{W} = \text{MIG}(\hat{I}), \tag{11}$$

where $I_i, \hat{I}_i \in \mathbb{R}^{H \times W}$. **Furthermore**, we obtain the semantic maps $S_i \in \{-1, 1, 2\}^{H \times W}$ of each $I_i$ using Grounded-SAM, where -1 refers to the background, 1 refers to $O_1$ and 2 refers to $O_2$.

## 4.2 MULTI-VIEW IMAGES TO 3D

We now have 1) the multi-view images and semantic maps $\{I_i, S_i\}_{i=1}^{W}$ of both $O_1$ and $O_2$, and 2) multi-view images $\{\hat{I}_i\}_{i=1}^{W}$ of only $O_1$. The target is to reconstruct 3D representations from these images and propagate the semantics from 2D images to 3D shapes. Since GroundedSAM does not guarantee multi-view consistent semantic segmentation, we leverage Part123 (Liu et al., 2024a) to integrate the multi-view semantic maps into a 3D consistent one. Specifically, Part123 optimizes a semantic aware NeuS (Wang et al., 2021) from $\{I_i, S_i\}_{i=1}^{W}$:

$$\{f, g\} = \text{Part123}(\{I_i, S_i\}_{i=1}^{W}), \tag{12}$$

where $f : \mathbb{R}^3 \mapsto \mathbb{R}$ is the SDF field of both $O_1$ and $O_2$, and $g : \mathbb{R}^3 \mapsto \mathbb{R}$ is the 3D semantic field. By marching cube algorithm, the mesh vertices can be extracted, denoted as $V = \{v_1, \ldots, v_N\}$. Then $V$

can be split into two groups $V = V_1 + V_2$ given the semantics from $g$. According to our assumption, $V_2$ denotes the mesh vertices of the foreground object $O_2$. We fit GS for both $O_1$ and $O_2$:

$$G_1 = \text{GaussianSplatting}(\{\hat{I}_i\}_{i=1}^W), \quad G_2 = \text{GaussianSplatting}(\{I_i\}_{i=1}^W; \mu \in V_2), \quad (13)$$

where we keep Gaussian centers of $G_2$ unchanged, i.e., based on the positions of $V_2$, to keep its surface consistent with the extracted SDF. The SDF is utilized as a boundary constraint for the following MPM simulation (Fuhrmann et al., 2003).

### 4.3 Physical Simulation-Enhanced Optimization

**Score Distillation Sampling Loss.** To ensure the generated Gaussian Splatting $G_1$ is semantically consistent with the description of the inpainted image, we adopt SDS (Poole et al., 2022) to further optimize its representation. The SDS loss is defined as:

$$\nabla_\theta \mathcal{L}_{SDS} = \mathbb{E}_{t,\epsilon}\left[ w(t)(\epsilon_\phi(I_t^p; y, t) - \epsilon)\frac{\partial I_t^p}{\partial \theta} \right], \quad (14)$$

where $w(t)$ denotes the time-dependent weighting function, $\epsilon_\phi$ represents the pre-trained 2D diffusion model, $I_t^p$ is the predicted image at timestep $t$. Here we reuse the text prompt $y$ for inpainting as the condition for generation. $\theta$ denotes the parameters of the target Gaussian Splatting representation i.e., $\{\mu, \Sigma, q, \alpha, c\}$ as mentioned in section 3.1. Among these parameters, $\mu$ represents the center position for each particle, which is the only key property to take care for ensuring physical plausibility. We freeze opacity $\alpha$ and color $c$ during the optimization to prevent SDS from changing the appearance of the object. Therefore we divided the parameters into three groups $\theta = \{\theta_\mu, \theta_t, \theta_a\}$, where $\theta_t$ denotes the scaling factor and rotation quaternion for the Gaussian particles. $\theta_a$ represents the frozen appearance-related parameters.

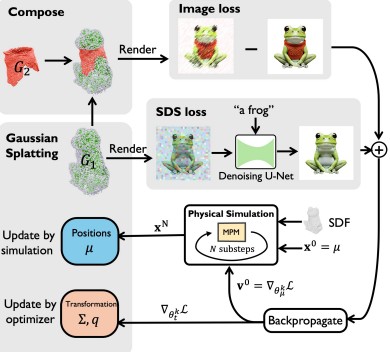

Figure 3: **The overview of our PSE-SDS**. We fix $G_2$ and optimize the attributes of $G_1$ to resolve the penetration issues caused by their direct composition. The gradients come from the SDS and image loss are divided into two streams during the backpropagation. Specifically, $\nabla_{\theta_\mu^k}\mathcal{L}$ is utilized as the initial velocity of the physical simulation for updating the positions $\mu$ of Gaussians.

**Image Loss.** As we constrain SDS loss for optimizing object geometry only, to ensure visual consistency, we utilize an image loss as a complement to penalize the L1-norm difference between the rendered from the generated composed object $G_c = \{G_1, G_2\}$ and the original input image:

$$\mathcal{L}_{Image} = (1 - \lambda_1)\mathcal{L}_1(I^c, I) + \lambda_1\mathcal{L}_{SSIM}(I^c, I), \quad (15)$$

where $I^c$ is the image rendered from the generated composed object, $I$ denotes the original input image, and $\mathcal{L}_{SSIM}$ refers to the structural similarity loss function.

**Physical Simulation-Enhanced SDS.** The final objective is to find parameters $\theta_\mu$ and $\theta_t$, by minimizing the total loss $\mathcal{L}$:

$$\mathcal{L} := \mathcal{L}_{Image}(\theta_\mu, \theta_t) + \lambda_3\mathcal{L}_{SDS}(\theta_\mu, \theta_t), \quad (16)$$

where the $\mathcal{L}_{SDS}$ and $\mathcal{L}_{Image}$ are designed to penalize discrepancy in geometry and visual appearance respectively, between the generated objects and the input image. $\lambda_3$ is the weighting factor.

We observed that directly applying the loss gradient to update particle positions $\mu$ results in penetrations and artifacts as shown in Figure 6. To ensure the physical plausibility, we propose physical simulation-enhanced SDS (shown in Figure 3). We delegate the updates of $\mu$ to the physical simulation. Here we use the MLS-MPM (Hu et al., 2018) as the physical simulator. One sub-step of the simulation process can be formalized as follows:

$$\mathbf{x}^{n+1}, \mathbf{v}^{n+1} = \mathrm{MPM}(\mathbf{x}^n, \mathbf{v}^n, \Delta t, \psi), \tag{17}$$

where $\mathbf{x}^n$ and $\mathbf{v}^n$ represent particle position and velocity at timestep $n$, $\psi$ denotes all other properties such as the particle mass, particle volume and materials parameters. Note we omit the subscript $p$ for clarity compared to the notations mentioned in section 3.2.

As described in algorithm 1, given $K$ steps of optimization, we set the $\nabla_{\theta_\mu^k}\mathcal{L}$ i.e., loss gradient with respect to particle position as the initial velocity of particles for the MPM based physical simulation. The MPM outputs the updated $\mu^{k+1}$ after $N$ sub-step simulations.

---

**Algorithm 1** Physical Simulation-Enhanced SDS

---

**Require:** Given $K$ steps of optimization, $N$ sub-steps MPM simulation, learning rate $\gamma$
1: **for** $k = 1$ to $K$ **do**
2:     Compute $\nabla_{\theta^k}\mathcal{L}$ according to Eqn.16
3:     $\nabla_{\theta^k}\mathcal{L} = \{\nabla_{\theta_\mu^k}\mathcal{L}, \nabla_{\theta_t^k}\mathcal{L}\}$
4:     $\mathbf{x}^0 = \mu^k, \mathbf{v}^0 = \nabla_{\theta_\mu^k}\mathcal{L}$
5:     $\Delta t = \gamma/N$
6:     **for** $n = 0$ to $N$ **do**
7:         $\mathbf{x}^{n+1}, \mathbf{v}^{n+1} = \mathrm{MPM}(\mathbf{x}^n, \mathbf{v}^n, \Delta t, \psi)$
8:     **end for**
9:     $\mu^{k+1} = \mathbf{x}^N$
10:    $\theta_t^{k+1} = \theta_t^k - \gamma\nabla_{\theta_t^k}\mathcal{L}$
11: **end for**

---

Intuitively, at the first sub-step of the simulation, the MPM advances the particles' positions according to the initial velocity (i.e., loss gradient), which is equivalent to one step of vanilla optimization using gradient descent with a step size $\Delta t$. The following simulation sub-steps are then performed to progressively correct the particles positions by solving the physical system.

## 5 EXPERIMENT

### 5.1 IMPLEMENTATION DETAILS

The prompts can be generated by a vision-language model (Team, 2025) or defined by the user. We use Stable-Diffusion-XL-1.0 as an inpainting model with a guidance scale in $\{7.5, 8.0, 9.0, 12.5\}$. During SDS optimization, we decrease timestep $t$ from 100 to 20. We train NeuS with 1k steps, fit $G_2$ with 30k steps and $G_1$ with 3k steps, and perform the physical simulation-enhanced optimization with 500 steps. We empirically set $\lambda_1 = 0.2, \lambda_2 = 1.0, \lambda_3 = 0.00001$.

### 5.2 EVALUATION

We assess the results with the following metrics: 1) Peak Signal-to-Noise Ratio (PSNR), which quantifies the similarity between the rendered image and the input image at the reference view; 2) CLIP score (Radford et al., 2021) for various comparisons, including between novel-view images and the input image ($\mathrm{CLIP}_{mv}$), between the reference view of $O_1$ and the inpainting prompt ($\mathrm{CLIP}_{text}$), between the reference view of $O_1$ and the inpainted image ($\mathrm{CLIP}_{ip}$), and between the novel-view images of $O_1$ and the inpainted image ($\mathrm{CLIP}_{ip}^{mv}$). 3) Penetration Rate (PR), which quantifying the proportion of points lying inside another component. We selected 20 images from the ComboVerse benchmark and generated additional 25 images using FLUX (Labs, 2024). These images feature

components with strong spatial coupling, such as "a cactus in a pot placed on a stool" and "a rhino wearing a large wool sweater".

Table 1: **Quantitative comparison with previous work.**

| Method | PSNR(dB)↑ | CLIP$_{mv}$(%)↑ | PR(%)↓ |
|---|---|---|---|
| Part123 (Liu et al., 2024a) | 22.68 | 82.01 | - |
| ComboVerse (Chen et al., 2024a) | 13.15 | 79.76 | 19.36 |
| Ours | **25.15** | **86.64** | **0.44** |

Figure 4: **Qualitative comparison with previous work.** The green box indicates decomposed objects, the orange box displays multi-view results, and the blue box highlights physical relationships (e.g., penetration, marked by red circles). Gaussian centers from the 3DGS representation are converted to point clouds for geometry visualization.

## 5.3 COMPARISON WITH BASELINE

We compare our approach with the following baselines: 1) Part123 (Liu et al., 2024a), which generates a holistic mesh with semantics from a single image; 2) ComboVerse (Chen et al., 2024a), which generates each component in the image separately, and assembles them with estimated similarity transformations. Fig. 4 and Tab. 1 shows the qualitative and quantitative results.

Overall, our method produces the most superior 3D compositional assets, taking into account both visual quality and physical plausibility. Part123 generates the entire assets as a single mesh, leading to incompletely segmented objects. ComboVerse can not address penetrations between objects. Our method achieves better consistency with the input image and effectively resolves the penetration problem. Note that Part123 performs semantic segmentation over a complete mesh, without providing individually modeled components. Consequently, it trivially reports a zero penetration rate, but this does not reflect a true compositional setting with physically independent parts. As such, a direct comparison is not meaningful for our use case.

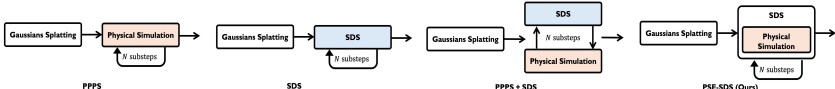

Figure 5: **Various methods of integrating interactive information through physical simulation**

## 5.4 ABLATION STUDY

We conduct the following ablation study to validate the effectiveness of our Physical Simulation-Enhanced SDS (PSE-SDS): 1) **PPPS** uses physical simulation as a post-processing procedure after the asset generation. 2) **SDS** denotes the vanilla SDS optimization that relying solely on visual supervision. 3) **PPPS + SDS** represents performing simulation and the vanilla SDS alternately. Figure 5 shows a comparison of how these variants integrate information through physical simulation.

**Effectiveness of Physical Simulation-Enhanced SDS.** The output generated in Stage 2 (Sec. 4.2) encounters penetration issues (indicated by red boxes in the second column of Fig. 6), due to the omission of interactive information in the process. Tab. 2 and Fig. 6 provide both quantitative and

Table 2: **Quantitative results of ablation studies on PSE-SDS.** The unit for PSNR is decibels (dB), and that for CLIP scores is percentage (%).

| Method | PSNR↑ | $\text{CLIP}_{text}$↑ | $\text{CLIP}_{ip}$↑ | $\text{CLIP}_{mv}$↑ | $\text{CLIP}_{ip}^{mv}$↑ |
|---|---|---|---|---|---|
| PPPS | 25.02 | **29.17** | **93.66** | 87.36 | 86.86 |
| SDS | 29.13 | 28.35 | 89.76 | 87.90 | 84.83 |
| PPPS + SDS | 20.74 | 28.17 | 89.46 | 88.06 | 84.69 |
| PSE-SDS (Ours) | **29.79** | 28.66 | 92.68 | **88.30** | **86.93** |

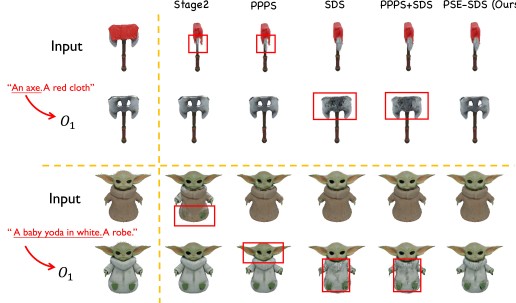

Figure 6: **Qualitative results of ablation studies on PSE-SDS.**

qualitative insights into the ablation studies examining various methods of integrating interactive information through physical simulation. 1) **PPPS** overlooks visual plausibility, since physical simulation treats every point as material without considering semantics. 2) **SDS** disregards physical plausibility; even though the overall asset aligns well with the input image, individual objects may collapse. 3) **PPPS+SDS** can still result in object collapse without adequate physical constraints. 4) Our **PSE-SDS** yields superior outcomes in terms of both visual and physical plausibility. We present more examples in Figure 8 to demonstrate that our method can generate assets with diverse compositional layouts.

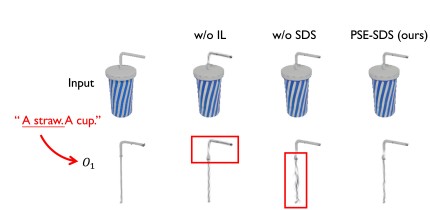

Figure 7: **Ablation studies on loss functions.**

Figure 8: **More results.**

**Are SDS and Image Loss both necessary?** We further assess the individual contributions of $\mathcal{L}_{SDS}$ and $\mathcal{L}_{Image}$, respectively (See Fig. 7). Excluding $\mathcal{L}_{Image}$ (w.o. IL) leads to the appearance of extraneous object, attributable to the variability inherent in SDS. Omitting $\mathcal{L}_{SDS}$ (w.o. SDS) results in poor visual plausibility within occluded areas.

## 6 CONCLUSION

In this paper, we present PhyCAGE, the first approach to generate physically plausible compositional 3D assets from a single Image. Our method incorporates a novel Physical Simulation-Enhanced Score Distillation Sampling (PSE-SDS) technique, which leverages a physical simulator as a physics-guided optimizer. This optimizer iteratively corrects the positions of the reconstructed Gaussians to achieve a physically compatible state. The experiments demonstrate that PhyCAGE is capable of generating various 3D assets in diverse compositional layouts. We believe our method represents a significant first step toward physics-aware 3D scene generation.

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

## A    RUNTIME COMPARISON

To summarize, using two components as an example, the main pipelines and runtime of the three methods are:

1. Part123 (Liu et al., 2024a): 332s
    (a) Single image to multi-view images: 51s

(b) Grounded-SAM on multi-view images: 17s

(c) Multi-view to NeuS: 264s

2. ComboVerse (Chen et al., 2024a): 757s

    (a) Grounded-SAM on single image: 1s

    (b) Inpainting for each component: $2 \times 3$s

    (c) Single image to multi-view for each component: $2 \times 51$s

    (d) Multi-view to NeuS for each component: $2 \times 264$s

    (e) SSDS optimization: 120s

3. Ours: 734s

    (a) Single image to multi-view images: 51s

    (b) Grounded-SAM on multi-view images: 17s

    (c) Inpainting for background object: 3s

    (d) Single image $\rightarrow$ multi-view for inpainted background: 51s

    (e) Multi-view to NeuS for foreground: 264s

    (f) Mesh to 3D Gaussian Splatting (3DGS) for foreground: 200s

    (g) Multi-view to 3DGS for background: 28s

    (h) PSE-SDS optimization: 120s

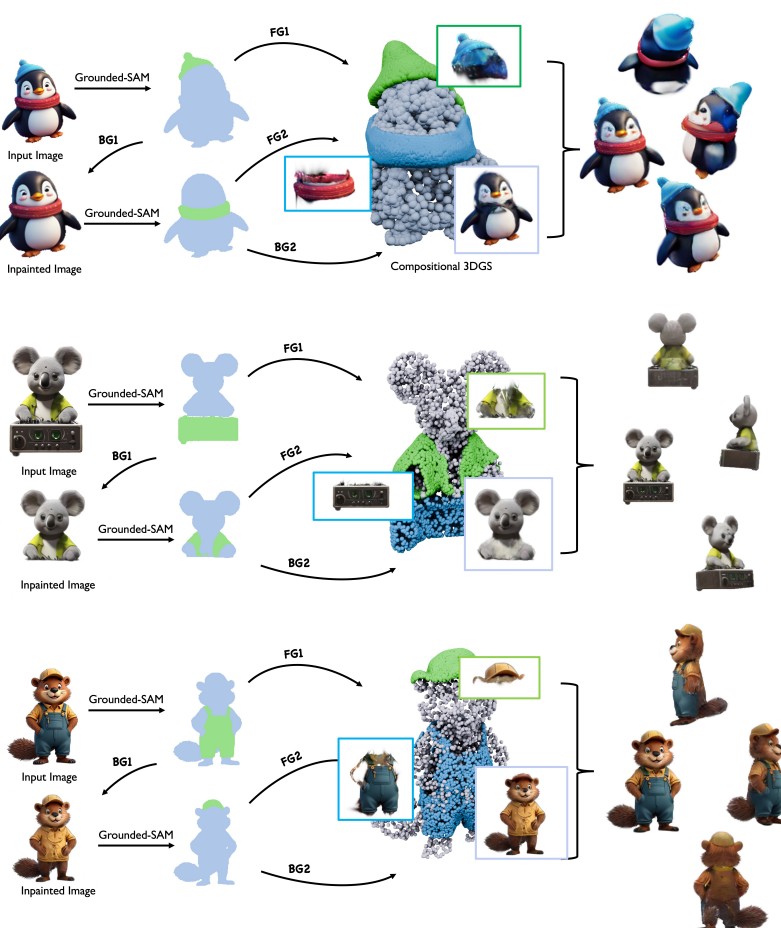

Figure 9: **Example scenario with multiple objects.** FG denotes foreground object and BG denotes background object.

## B  PENETRATION RATE COMPUTATION

We evaluate our mothod with the baselines using penetration rate to demonstrate the effectiveness of our method in reducing inter-object penetration.

A common approach for quantifying mesh-based penetration is to sample points on one mesh and compute the proportion of points lying inside another mesh. While this is applicable to mesh-based approaches like ComboVerse, our method is based on 3D Gaussian Splatting (3DGS) from sparse-view inputs, which makes surface extraction non-trivial.

Instead, we leverage the Signed Distance Field (SDF) used in our simulation and quantify penetration as the proportion of points that lie inside other objects—i.e., those with positive SDF values.

## C  SCENARIO WITH MULTIPLE OBJECTS

Figure 9 presents a case involving multiple objects. We iteratively designate one object as the foreground while treating the remaining objects as the background within each generation sub-routine.

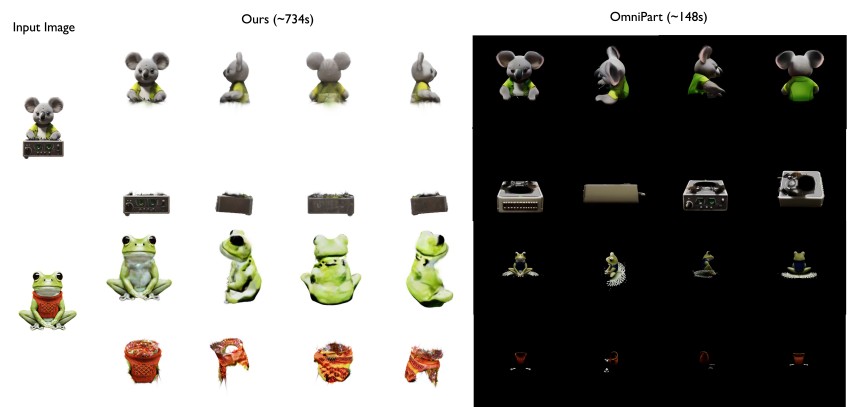

Figure 10: **Comparison with OmniPart (Yang et al., 2025).**

## D  COMPARISON WITH DIFFUSION-BASED METHODS

Diffusion-based methods (Lin et al., 2025; Yang et al., 2025) in compositional 3D generation has recently gained popularity. PartCrafter (Lin et al., 2025) is a structured 3D generative model that jointly synthesizes multiple semantically meaningful and geometrically distinct 3D meshes from a single RGB image. However, it only focus on geometry and does not support custom masks input. OmniPart (Yang et al., 2025) generates compositional 3D assets from an image and the corresponding mask input by first planning a 3D part structure using an autoregressive model and then synthesizes all high-quality, textured parts simultaneously.

Diffusion-based methods typically rely on large-scale data-driven training. Their faithfulness to the input image depends heavily on the model's generalization ability and robustness. The advantage is that they can produce results in a single diffusion pass, making them very fast.

In contrast, our method focuses on an optimization-based paradigm. It does not require large training datasets, and the alignment between the generated 3D model and the input image can be progressively improved through optimization. However, SDS-based optimization is computationally slower and less stable. We include several qualitative examples in Fig.10 to illustrate these characteristics.

## E  LIMITATIONS AND FUTURE WORK.

The quality of the final output depends on the performance of both the multi-view generation method and the sparse-view 3D Gaussian splatting technique. We expect that our approach can be further improved by leveraging more robust reconstruction model in the future. We aim to further develop our method to generate mesh-based assets, thereby supporting more simulation techniques such as the Finite Element Method (FEM) (Sifakis & Barbic, 2012) and sophisticated collision handling methods (Li et al., 2020).

## F  USAGE OF LLM USAGE

We use GPT to polish our writing.

## G  ADDITIONAL EXPERIMENT

In this section, we present extended experimental results comparing 3D model generation from diverse input images using our proposed method and ComboVerse (Chen et al., 2024a).

As demonstrated in Fig. 12 and Fig. 13, our approach consistently generates 3D assets with superior quality, greater diversity, and enhanced multi-view consistency compared to the baseline method.

Furthermore, Fig. 11 provides geometric analysis of our generated compositional assets, where we visualize the underlying structure by converting 3D Gaussian Splatting (3DGS) centers into point cloud representations. This visualization highlights the physical plausibility achieved by our method.

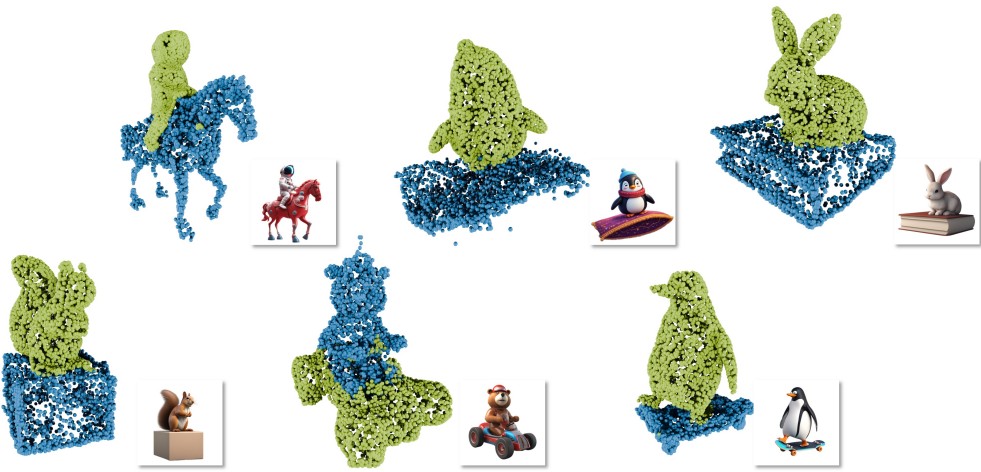

Figure 11: **Additional Geometric Visualizations.**

Input Image                                    Generated models

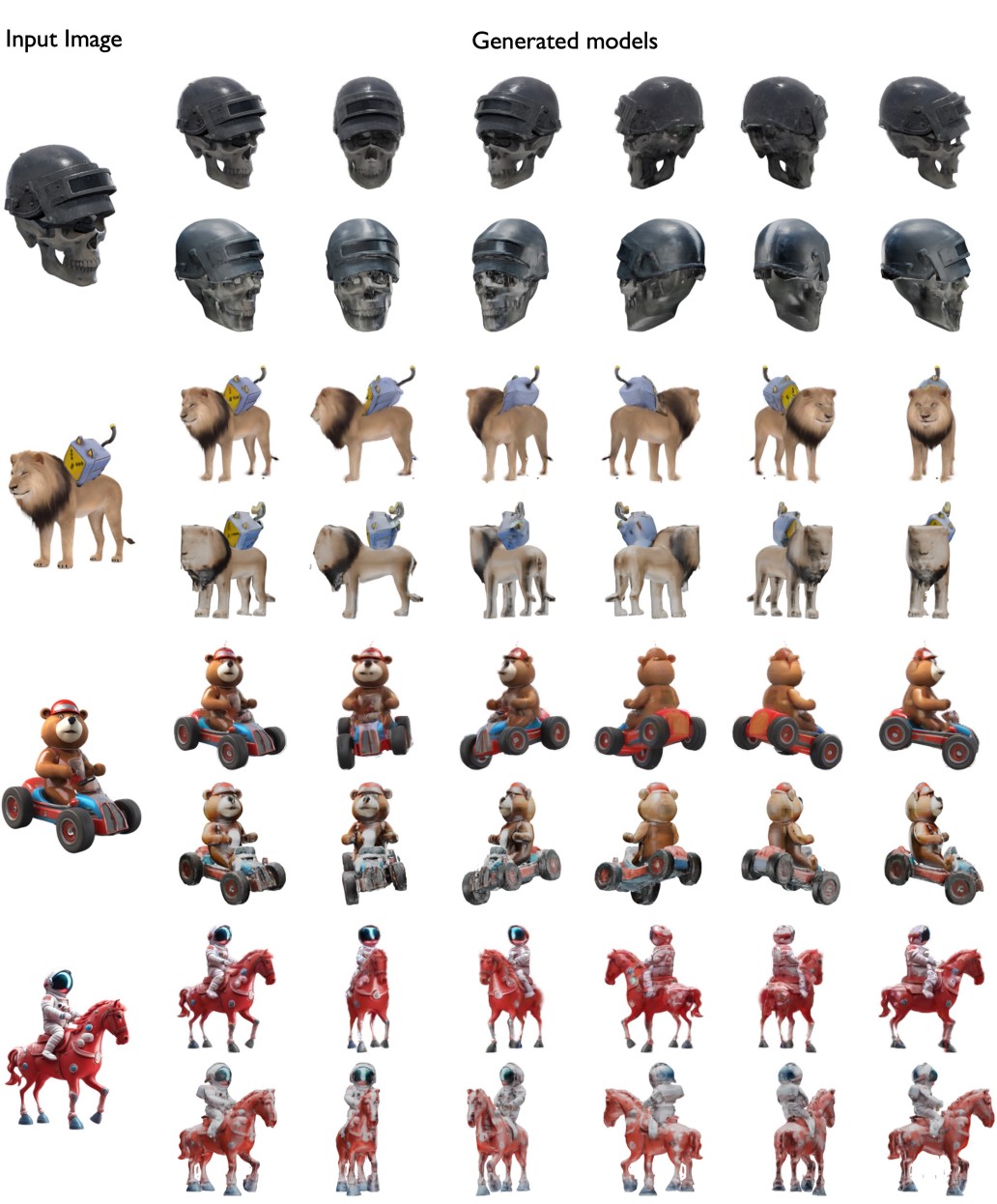

Figure 12: **Qualitative comparison with Comboverse.** For each example, the top row presents our method's results, while the bottom row displays ComboVerse's outputs.

Input Image                          Generated models

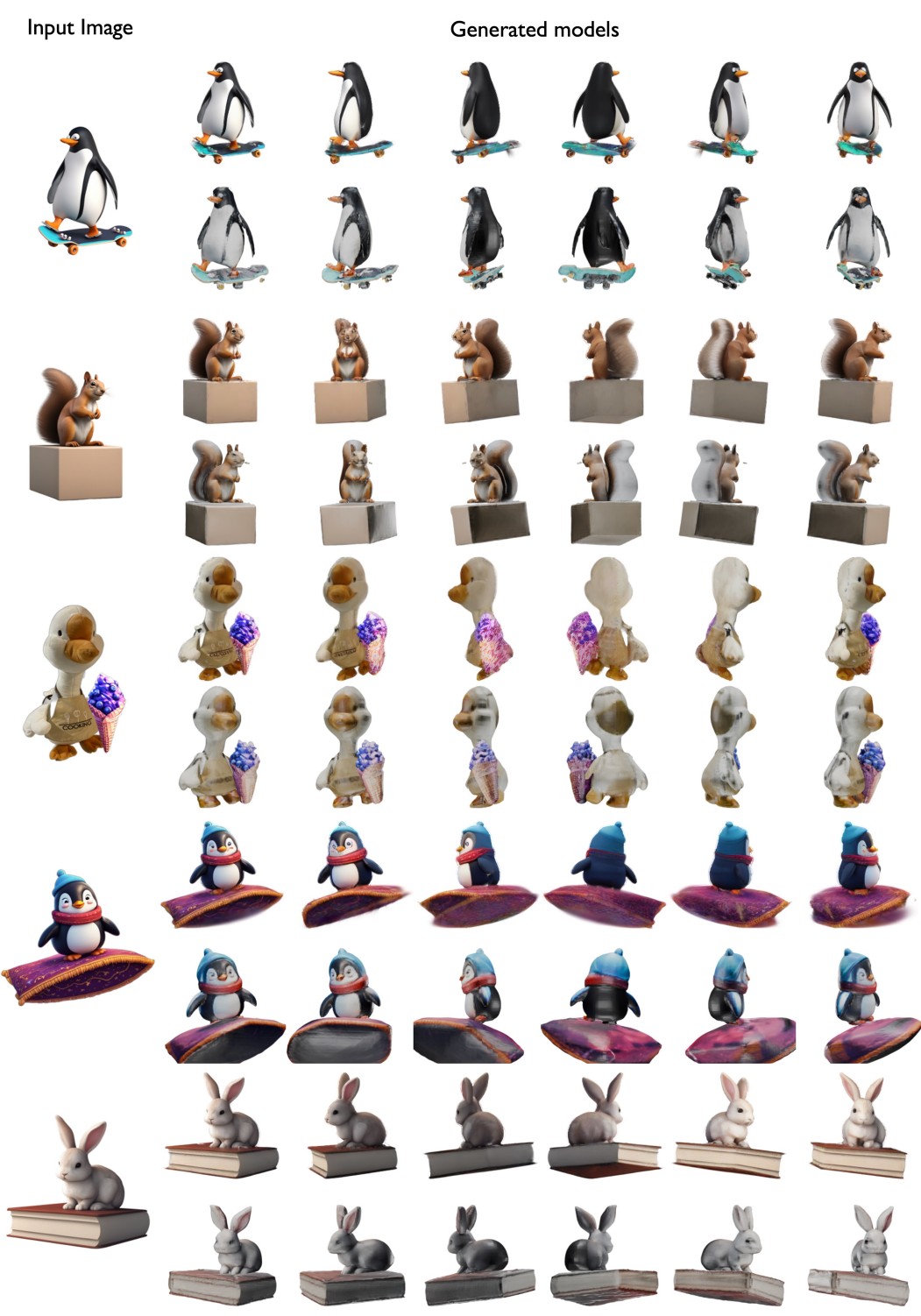

Figure 13: **Qualitative comparison with ComboVerse.** For each example, the top row presents our method's results, while the bottom row displays ComboVerse's outputs.

