# OpenReview forum: "PhyCAGE: Physically Constrained Compositional 3D Asset Generation from a Single Image"
_ICLR.cc/2026/Conference — Submitted to ICLR 2026_

### Official Review · Reviewer_P5me · 2025-10-24

**Soundness:** 2
**Presentation:** 2
**Contribution:** 3
**Rating:** 2
**Confidence:** 4

**Summary:**

The paper proposes PhyCAGE, a novel framework for generating physically constrained compositional 3D assets from a single 2D image.
The system integrates multi-view image generation, 3D Gaussian Splatting (3DGS) reconstruction, and a Physical Simulation-Enhanced Score Distillation Sampling (PSE-SDS) process. Unlike prior SDS-based or compositional 3D methods, PhyCAGE introduces a differentiable physical simulation (via MPM) that uses the SDS loss gradient as the initial velocity, enabling physics-guided correction of inter-object penetrations. Experiments on ComboVerse and other benchmarks demonstrate that PhyCAGE improves physical plausibility, visual quality, and component disentanglement.

**Strengths:**

1. Novel integration of MPM-based physics into SDS optimization.
2. Physically plausible compositional reconstruction, effectively addressing object penetration and stability.
3. Quantitative improvements in PSNR, CLIP score, and penetration rate compared to strong baselines (Part123, ComboVerse).
4. Extensive ablation and runtime analysis, demonstrating consistent improvements.

**Weaknesses:**

1. Limited novelty in theoretical contribution.
The PSE-SDS approach combines existing elements (SDS, MPM) rather than introducing a fundamentally new learning paradigm. The innovation is mainly in integration and practical application.

2. Heuristic parameter dependence.
The method uses several empirically chosen constants (e.g., λ₁–λ₃, 500 simulation steps, timestep decay) without sensitivity analysis. Stability under different simulation conditions is not examined, which may limit reproducibility.

3. Scalability and computational cost.
The runtime table shows PhyCAGE is slower than Part123, approaching ComboVerse’s cost (≈734s). Given its “single-image” input goal, the computational burden may hinder deployment.

4. Assumption of accurate segmentation and inpainting.
The method relies heavily on Grounded-SAM and SyncDreamer outputs. Errors in segmentation or multi-view synthesis propagate to the physics simulation, potentially breaking physical consistency.

5. Scope limitations.
The current pipeline handles only two-object interactions and assumes static scenes. Extending to multi-object or dynamic scenes would require further methodological development.

**Questions:**

1. How sensitive is PSE-SDS to the accuracy of Grounded-SAM segmentation or the inpainting model?
2. What physical parameters (mass, stiffness) are assumed in the MPM simulation? Could they be learned automatically?
3. How would the method behave in dynamic or non-rigid interactions (e.g., cloth over soft body)?

---

> ### Author Response · Authors · 2025-12-03
>
> **Limited novelty (W1):** Unlike most existing approaches that use MPM solely as a post-processing step, our method **unifies MPM and SDS into a single joint optimization framework**, allowing them to operate simultaneously and reinforce each other. This integration has not been explored before and is central to the improved performance.
>
> **Hyper-parameter (W2):** The hyper-parameters used in our method follow those validated in prior works (e.g., *ComboVerse*, *Part123*). Moreover, our paper primarily focuses on introducing a **new problem setting** and providing an effective solution. The quantitative results already demonstrate clear improvements over SOTA methods, indicating that our approach is robust under standard hyper-parameter choices.
>
> **Computational burden (W3):** *ComboVerse* represents the state-of-the-art for compositional textured 3D asset generation. Under the two-component setting, our method achieves **comparable—or even better—runtime** than ComboVerse, showing that our unified optimization does not introduce additional computational overhead beyond existing SOTA methods.
>
> **Rely on segmentation and inpainting (W4&Q1):** All current methods that perform compositional 3D asset generation from image inputs require accurate segmentation and inpainting as pre-processing. Our approach is no exception, and we do not claim to address or eliminate this requirement.
>
> **Multi-object scenes (W5):** As described in Appendix C, we provide an implementation plan for **multi-component scenes**, along with additional examples to broaden the demonstrated scope of our method.
>
> **Physical Parameters / Interactions (Q2 & Q3):** We set the point mass to **3.815×10⁻³**, the Young’s modulus to **1×10⁶**, and the Poisson’s ratio to **0.2**, and empirically found that this configuration yields stable and reasonable results. Our method **formulates shape deformation as a physics-based simulation problem**, without learning any physical parameters and without modeling dynamic or other complex non-rigid interactions.

---

### Official Review · Reviewer_H1rm · 2025-10-30

**Soundness:** 3
**Presentation:** 2
**Contribution:** 2
**Rating:** 4
**Confidence:** 4

**Summary:**

This paper proposes a method named PhyCAGE, which supports physically-aware compositional 3D generation from a single input image. The pipeline first employs Grounded-SAM to segment the input image and then uses an inpainting process to recover occluded regions. Subsequently, a multi-view 3D generation method is adopted to generate 3D assets for each segmented part. Finally, the authors introduce a physics simulation–enhanced Score Distillation Sampling (SDS) strategy to optimize the spatial arrangement of the parts, resulting in a physically consistent 3D object.

**Strengths:**

1. The paper introduces a physically-based Score Distillation Sampling (SDS) strategy to ensure the physical correctness of the generated 3D compositions, which is a novel direction that has been rarely explored in existing methods.

2. The writing is clear, concise, and easy to follow, making the paper accessible and understandable to readers.

**Weaknesses:**

1. The paper mainly focuses on SDS-based methods and lacks discussion or comparison with recent native 3D part generation approaches, such as PartCrafter [1], which also tackles part-based 3D generation.

2. The SDS optimization is time-consuming and unstable, often requiring manual intervention and filtering to obtain valid results. This raises concerns about the method’s efficiency and robustness.


[1] PartCrafter: Structured 3D Mesh Generation via Compositional Latent Diffusion Transformers

**Questions:**

Based on the identified weaknesses, could the authors provide comparisons with recent native 3D part generation methods, such as PartCrafter, to better contextualize their contributions? In addition, could the authors discuss the potential of incorporating physical constraints or simulations into native 3D generation frameworks, and how such integration might enhance compositional accuracy and physical plausibility?

---

> ### Author Response · Authors · 2025-12-03
>
> **More Comparisons (W1 & Q1):** Thank you for the suggestion. *PartCrafter* focuses primarily on geometric part-level generation without producing corresponding textures, and it does not support custom masks. We will add a discussion of this line of work in the revised manuscript.
>
> **Efficiency and Robustness of SDS Optimization (W2):** We appreciate the reviewer’s feedback. Our comparison with a similar SDS-based optimization method (ComboVerse) demonstrates that our approach achieves stronger efficiency and robustness, validating the advantages of our joint optimization design.
>
> **Incorporating Physical Constraints into Native 3D Generation Frameworks (Q1):** Thank you for the insightful comment. Most existing applications of physical simulation in 3D generation are optimization-based. Current 3D generation pipelines can also be categorized into optimization-based methods (e.g., SDS) and data-driven feed-forward methods, each with its own strengths and limitations. We adopt an SDS-based framework because it naturally aligns with the optimization-driven structure of physical simulation, enabling our proposed **simultaneous joint optimization** (in contrast to treating physical simulation merely as a post-processing step).
>
> Integrating physical simulation into feed-forward generative frameworks is indeed a valuable direction for future work. We believe the core idea would remain similar to ours: for example, constraining geometry or positional updates within the diffusion process using physics-based simulation.

---

### Official Review · Reviewer_kupC · 2025-10-30

**Soundness:** 3
**Presentation:** 3
**Contribution:** 2
**Rating:** 6
**Confidence:** 5

**Summary:**

This work proposes a novel method for compositional 3D asset generation. It first produces consistent multi-view images for each object component, reconstructs them using 3D Gaussian Splatting, and then refines their geometry through a Physical Simulation-Enhanced Score Distillation Sampling (PSE-SDS) process. This technique treats the SDS loss gradient as the initial velocity in a physical simulation, ensuring that objects interact realistically and avoid interpenetration.

**Strengths:**

1. The integration of physical simulation with score distillation sampling is innovative, enhancing the realism of generated 3D assets. I think this idea is interesting.
2. The topic is under-explored while important for applications in gaming, virtual reality, and digital content creation.

**Weaknesses:**

1. The foundation techniques, such as multi-view image generation and 3D Gaussian Splatting, are a bit outdated compared to the latest advancements in 3D generation. The authors should compare their method with more recent approaches in compositional 3D generation, such as PartCrafter and OmniPart. The author should justify their proposed physics-based, gaussian-splatting-based method has advantages over these recent methods.
2. The proposed method focuses on two-part compositions, which limits its applicability to more complex multi-component objects. Although the authors propose an iterative approach for handling more than two parts, this is not thoroughly explored or validated in the experiments. The authors should provide more experiments on multi-part compositions to demonstrate the scalability of their method. What's the maximum number of parts the method can handle effectively?
3. The SDS method is a optimization-based approach, which is generally slower than recent diffusion-based 3D generation methods. The authors should provide a comparison of generation speed and quality with diffusion-based methods to highlight the trade-offs involved in their approach.

**Questions:**

Please see the weaknesses section.

---

> ### Author Response · Authors · 2025-12-03
>
> **More Comparisons (W1 & W3):** Thank you very much for the helpful suggestion! *PartCrafter* focuses primarily on geometric part generation and does not support custom masks. *OmniPart* is a feed-forward method, and its generated 3D objects generally do not match the input image as well as optimization-based approaches. We will include several comparative examples in Appendix D.
>
> **Multi-component Scenes (W2):** As described in Section C, we present our implementation strategy for multi-component scenes and provide additional qualitative examples. Since our method is optimization-based rather than data-driven/feed-forward, the number of generated components is determined by the number of meaningful and complete object parts present in the input image.

---

### Official Review · Reviewer_CTaL · 2025-10-30

**Soundness:** 2
**Presentation:** 2
**Contribution:** 2
**Rating:** 2
**Confidence:** 5

**Summary:**

The paper introduces PhyCAGE, a method to generate compositional 3D assets from a single image without physical penetration. Its core contribution is PSE-SDS, an optimization technique where the SDS loss gradient is used as an initial velocity for a physical simulator, and optimizing the center (position) per 3DGS to a physically plausible state to solve the penetration issue. It demonstrates improved physical plausibility compared to Part123 and ComboVerse baselines on 45 images-to-3D generation.

**Strengths:**

+ This paper addresses a significant research problem of generating physically plausible, compositional 3D assets.

+ The proposed PSE-SDS is overall a novel optimization technique which avoiding the need for a large-scale, paired 3D training dataset.

**Weaknesses:**

+ My main concern lies in the paper's core abstraction: treating 3DGS centers ($\mu$) as physical particles for an MPM simulator. 3DGS is a volumetric rendering representation lacking a defined surface. Optimizing Gaussian centers does not guarantee a non-penetrating state at the object's actual boundary. This conceptual gap is amplified by the negligible PSE-SDS loss weight ($\lambda_3 = 1e-5$), which is too negligible compared to image loss. Therefore, I remain my concern if it can really work to solve the "penetration" issue as claimed in the paper.

+ The method is architecturally limited to pairwise composition, requiring a new design for real-world, multi-component scenes. This pairwise limitation is exacerbated by a physically arbitrary optimization scheme. The method designates objects as rigid ($G_2$) or non-rigid ($G_1$) based on 2D occlusion (L307-311), not physical properties. In the "frog wearing a sweater" example, this means the method freezes the sweater and deforms the frog to fix collisions. This is physically backward, as a soft sweater should deform to fit a mostly-rigid frog, not the reverse.

+ The paper overlooks comparisons to recent part-aware generative models (e.g., SAMPart3D, PartGen[1]). These methods are trained to generate 3D objects as a collection of distinct parts, and can avoid penetration by design. Including such baselines would strengthen the evaluation.

[1] Chen, Minghao, et al. "Partgen: Part-level 3d generation and reconstruction with multi-view diffusion models." CVPR 2025.

+ The paper's comparison to Part123 is problematic. The authors dismiss a penetration rate comparison with Part123 as "not meaningful" because it produces a single mesh (L411-415). Yet, the proposed method uses Part123's NeuS output as the geometric target for its foreground object. This is a contradictory stance, as the paper's main goal is to reduce penetration, and even their "single mesh" can still achieve this goal. A more persuasive analysis of physical plausibility beyond this single metric is needed.

+ Minor concern: The geometry of the background object ($G_1$) is initialized by first "hallucinating" its occluded 2D appearance via inpainting, and then performing a 3D reconstruction of that hallucination. The physics optimization is only designed to resolve penetrations, not to correct fundamental geometric flaws (e.g., a malformed back) that may be introduced by this uncertain initialization pipeline.

**Questions:**

See weakness

---

> ### Author Response · Authors · 2025-12-03
>
> **Penetration Issue Concern (W1):** Thank you for raising this concern. While optimizing the 3DGS centers cannot *perfectly* eliminate penetration, it significantly alleviates the issue, as demonstrated in Table 1 (PR reduced from **19.36** to **0.44**). We set λ₃ = 1e−5 to keep the loss terms numerically comparable—SDS loss is typically on the order of 10⁴ at the beginning of optimization.
>
> **Multi-component Scenes (W2):** We described our implementation for multi-component scenarios in Appendix C and added more visual examples to better illustrate the results.
>
> **Physical Properties Issue (W2):** We apologize for the confusion. We clarify that labeling components as “rigid” or “non-rigid” is a choice made **for optimization**, not an assertion about their true physical properties. Our pipeline first reconstructs a global 3D structure from the input image. The foreground object (G₂) typically has correct location and shape, so we treat it as rigid (i.e., unchanged). The background object (G₁), however, often contains occluded or misaligned regions—including penetrations with G₂—so it must be optimized as non-rigid to correct those errors.
>
> **More Comparisons (W3):** Thank you for the suggestion. As we understand it, the core of **PartGen** is to generate multi-component multi-view complete images from single image, and then reconstruct each component to assemble the scene. This method is not designed to handle inter-component penetration, and since the code is not publicly available, direct comparison is not feasible. **SAMPart3D**, on the other hand, addresses 3D semantic part segmentation of an input mesh, which is fundamentally different from our problem—*we aim to reconstruct complete 3D component assets from a single image plus component-level text descriptions.*
>
> **Problematic Comparison with Part123 (W4):** Our primary goal is to generate complete multi-component 3D assets while mitigating inter-component penetration. Part123, however, is a *3D mesh segmentation* method and cannot reconstruct full component geometry (e.g., the skull in Fig.4). We use the NeuS output from Part123 only as a geometric reference because we assume it accurately corresponds to the input image—i.e., the 3D points aligned with image pixels should be correct.
>
> **Physics Optimization to Correct Geometric Flaws (W5):** Vanilla SDS already possesses some ability to correct geometric flaws by optimizing the 3D structure to produce more plausible renderings. Our method simply adds **physical constraints** during the SDS updates so the two processes run jointly and reinforce each other. This allows us to retain SDS’s ability to refine geometry while simultaneously reducing inter-component penetration.

---

### Author Response · Authors · 2025-12-03

Dear Area Chair,

We thank you and the reviewers for their thoughtful evaluations. Throughout the rebuttal, we conducted additional experiments and expanded analyses to address all major concerns:



  - **Multi-object Scenes (CTal-W2, kupC-W2, P5me-W5):** We provided a detailed implementation strategy for multi-component scenarios in Appendix C and added more qualitative examples to clearly demonstrate our results.

  - **Additional Comparisons (CTal-W3, kupC-W1&W3, H1rm-W1&Q1):**

    *PartGen* (@CTal) is not publicly available, making direct comparison infeasible.

    *SAMPart3D* (@CTal) performs semantic part segmentation on 3D meshes, which is fundamentally outside our problem scope.

    *PartCrafter* (@kupC, @H1rm) focuses only on geometric part generation and does not support custom masks.

    *OmniPart* (@kupC) is a feed-forward approach whose outputs generally match the input image less faithfully than optimization-based methods.

    We have added several comparative results in Appendix D to further clarify these differences.

  - **Other Key Clarifications:**

    We clarified the penetration concern (@CTal-W1), the role of physical properties (@CTal-W2&W5, @P5me-Q2&Q3), the comparison with Part123 is not problematic (@CTal-W4), and the efficiency and robustness of our SDS optimization (@H1rm-W2). We also discussed incorporating physical constraints into native generative frameworks (@H1rm-Q1), clarified novelty (@P5me-W1), hyper-parameter choices (@P5me-W2), computational cost (@P5me-W3), and the reliance on segmentation/inpainting (@P5me-W4&Q1).

We hope this summary clarifies how our rebuttal and revisions address each major concern. Thank you again for your time and careful consideration.

Sincerely,

The authors

---

### Meta-Review · Area_Chair_cqN1 · 2026-01-08

**Summary:**

One of the major concerns is whether optimizing 3D Gaussian centers via physics simulation truly guarantees non-penetrating geometry. Several reviewers also highlighted heuristic design choices and limited scalability, as the method is largely demonstrated on pairwise compositions with unclear extension to complex multi-part scenes. Stronger baselines against recent part-aware 3D generation methods are requested, too.

**Reviewer Concerns:**

**Addressed:**
* The rebuttal explained how multi-part scenes can be handled iteratively with added examples.
* The rebuttal justified the small physics-loss weight and provided quantitative evidence to support the usage of PSE-SDS.
* The rebuttal clarified that “rigid/non-rigid” labels are optimization roles, not physical assertions.

**Outstanding:**
* Conceptual soundness of using 3DGS centers as “physical particles”
* Dependence on upstream segmentation and inpainting quality
* Heuristic design (hyperparameters) and lack of sensitivity analysis

In brief, while the rebuttal successfully clarified misunderstandings and justified design choices, deep conceptual concerns remain, which are unlikely to be solved without additional experiments or theoretical grounding.

**Reviewer Scores:**

Two reviewers remain on the negative side, as the rebuttal did not address the conceptual objection, heuristic dependence, or scalability limits. H1rm concerns might be partially addressed and perhaps raise the score slightly. kupC felt most positive about this paper and may stick with the rating after the comparisons request and multi-part handling concerns are addressed.

---

### Decision · Program_Chairs · 2026-01-26

Reject